# Towards Understanding How Machines Can Learn Causal Overhypotheses

## Abstract

Recent work in machine learning and cognitive science has suggested that understanding causal information is essential to the development of intelligence. One of the key challenges for current machine learning algorithms is modeling and understanding *causal overhypotheses*: transferable abstract hypotheses about sets of causal relationships. In contrast, even young children spontaneously learn causal overhypotheses, and use these to guide their exploration or to generalize to new situations. This has been demonstrated in a variety of cognitive science experiments using the "blicket detector" environment. We present a causal learning benchmark adapting the "blicket" environment for machine learning agents and evaluate a range of state-of-the-art methods in this environment. We find that although most agents have no problem learning causal structures seen during training, they are unable to learn causal *overhypotheses* from these experiences, and thus cannot generalize to new settings.

## 1 Introduction

Over the past few years, deep learning has made impressive progress in many areas, including reinforcement learning, natural language processing, and computer vision. However, most state-of-the-art algorithms still pale in comparison to humans, even children, for out-of-distribution generalization and fast adaptation to new tasks. In contrast, causal modeling has long concerned itself not merely with accurate modeling of in-distribution data, but also the accurate recovery of underlying causal mechanisms (and their true graphical relations) capable of explaining *out-of-distribution* data. Hence causal modeling holds promise for achieving systematic generalization (Bengio et al., 2019; Schölkopf et al., 2021; Ke et al., 2021).

One of the key components of causal learning for humans are causal overhypotheses, which describe priors over causal graphs. For example, a causal hypothesis may be that a system is stochastic, or that it is *conjuntive*—i.e., more than one cause must be present in order for the effect to occur. Causal overhypotheses are important because they enable humans to learn causal models from a sparse amount of data (Griffiths & Tenenbaum, 2009), by reducing the set of possible causal relationships to consider. Despite the recent surge of machine learning datasets and environment for causal inference and learning (Ahmed et al., 2020; McDuff et al., 2021; Wang et al., 2021a; Ke et al., 2021), the causal overhypotheses for these environments and datasets are unclear. Thus we cannot use these existing benchmarks to evaluate how capable are agents are at learning and using causal overhypotheses.

In this work, we seek to fill this gap, by focusing on introducing a benchmark environment that draws inspiration from recent cognitive science work using blicket detectors. This environment contains a virtual blicket detector, and has been used previously for understanding causal learning and exploration in children (Kosoy et al., 2022) and basic RL models, but not yet for state-of-the-art machine learning agents.

A "blicket detector" is a machine that lights up and plays music when some combinations of objects but not others are placed on it (Gopnik & Sobel, 2000; Lucas et al., 2014). The central question is whether an agent can learn that a particular set of causal events will lead to the lighting-up effect, and use that knowledge to design novel interventions on the machine. The causal relationship is entirely determined by the pattern of conditional dependencies and interventions, rather than requiring intuitive physics knowledge or visual understanding. Although these tasks may seem simple, and are easily mastered by children, we find that they are challenging for current learning algorithms.

Several features of this environment and the tasks it allows make it particularly useful as a benchmark for machine learning algorithms. First, causal representations are more powerful and structured than mere statistical generalizations, though both can be systematically inferred from statistical information. Many researchers (e.g. Pearl, Spirtes et al., Bengio) have argued that such causal representations are crucial for both human and general AI intelligence. The goals of causal inference, namely learning which actions will alter the environment in particular ways, are similar to those of standard reinforcement learning, but adding causal representations and inferences makes such learning far more effective.

Second, and unlike some existing causal environments (Ke et al., 2021; Wang et al., 2021a), the blicket environment enables the inference of both *specific causal structure* and more *general features of causal structure*, such as whether causal systems are conjunctive or disjunctive, stochastic or deterministic. Learning these *overhypotheses* about causal structure (Griffiths & Tenenbaum, 2009) is especially important because such inferences can constrain the search for causal structure in the future, a search that can rapidly become unwieldy. Third, this environment allows for active learning and exploration in a way that is both sufficiently constrained to be practical and that also allows for informative interventions.

Most significantly, and again unlike existing environments, research has already shown that even preschool children can easily manipulate and explore this environment, generate appropriate data, and rapidly learn both particular causal structure and overhypotheses about causal structure (Kosoy et al., 2022). We can then directly compare both the overall performance and the behavior of various agents in these tasks to the performance and behavior of children. Young children are a particularly informative baseline group. They do not have the extensive education and experience of typical adults, which might make comparisons to artificial agents challenging, but they are nevertheless effective causal learners and able to make broad yet accurate generalizations from small sample sizes, in contrast to many current machine learning systems (Gopnik, 2012; Gopnik et al., 2017).

The work of Kosoy et al. (2022) first described a version of this environment and included data on children and a few learning algorithms. But the environment and benchmarks have much wider applications than in that preliminary study or the studies reported here. It allows multiple comparisons of children and agents on multiple tasks and allows direct comparison with ML systems for multiple kinds of causal inference tasks. In Kosoy et al. (2022) and in the experiments described here we focus on one such inference – to conjunctive versus disjunctive structure, and a few algorithms. But with very minor changes the environment would allow tests of inferences to many other kinds of overhypotheses and inductive biases, such as inferring whether systems are stochastic or deterministic or require sequential or unordered interventions, overhypotheses that are also important for causal inference. It also allows multiple measures of causal inference including interventions and counterfactuals as well as predictions. And, significantly, it would allow researchers to empirically record the exploration behavior that children use in solving causal problems, not yet done this in this paper, and compare it to a wide variety of increasingly influential exploration based ML procedures.

In what follows, we look at a spectrum of algorithms ranging from reinforcement learning, to imitation learning, to the use of pre-trained language models, to better understand how these different methods perform on the proposed environment and tasks. We find that such algorithms, in contrast to children, only converge on a solution after an extensive number of trials or if they are given all the possible settings and outcomes as training data. This suggests that these tasks are an interesting challenge for machine learning algorithms. In order for machines to perform as well as children do, algorithms must reason about the sequence of observations seen, extract causal overhypotheses from those observations and use them for exploration—which current methods fall short of doing.

## 2 RELATED WORK

**Exploration in Reinforcement Learning** Recent exploration algorithms for deep reinforcement learning typically add an exploration bonus to the task reward; please refer to Amin et al. (2021) for a comprehensive survey. This bonus could be based on novelty (Bellemare et al., 2016; Ostrovski et al., 2017; Martin et al., 2017; Tang et al., 2017; Machado et al., 2018a), dynamics prediction error (Schmidhuber, 1991; Pathak et al., 2017), uncertainty (Osband et al., 2016; Burda et al., 2018), or disagreement (Pathak et al., 2019). The vast majority of existing exploration methods do not involve causal reasoning, whereas our proposed environment requires algorithms to learn and use causal overhypotheses in order to explore effectively, to solve the task.

**Multi-task and Causal RL Benchmarks** There exist multi-task RL benchmarks featuring robotics (Yu et al., 2019; James et al., 2020), physical reasoning (Bakhtin et al., 2019; Allen et al., 2020), and video games (Cobbe et al., 2018; Machado et al., 2018b; Nichol et al., 2018; Chevalier-Boisvert et al., 2018). Unfortunately, since it is not clear what the relevant causal overhypotheses for these environments are, it is difficult to evaluate how causal information influences agents' exploration.

RL benchmarks for causal discovery include Causal World (Ahmed et al., 2020), Causal City (McDuff et al., 2021), Alchemy (Wang et al., 2021a), ACRE (Zhang et al., 2021), and the work of Ke et al. (2021). However, many of these environments either lack clear causal hypotheses or do not allow for controlling overhypotheses. In addition, these environments primarily focus on causal induction or generalization, rather than exploration (though see Sontakke et al., 2021). In contrast, the blicket environment in this work is designed to measure agents' ability to explore using causal overhypotheses. Moreover, children have not been tested on any of these existing environments, whereas in the blicket environment, prior work has shown that children as young as age four are able to learn causal overhypotheses and use these to explore effectively (Kosoy et al., 2022). It can be informative to compare the exploration and performance of RL approaches to that of children.

**Language Models for Reasoning Tasks** Large language models such as GPT (Radford et al., 2019; Brown et al., 2020) and PALM (Chowdhery et al., 2022) are trained on massive amounts of data, and they have been shown to be able to express uncertainty and perform common sense reasoning up to an extent (Lin et al., 2022). In this work, we probe the the causal reasoning capabilities of GPT-3 and PALM using textual descriptions of the virtual blicket environment.

## 3 THE "BLICKET DETECTOR" ENVIRONMENT

The blicket detector is commonly used in cognitive science to study causal inference and reasoning in children (Gopnik & Sobel, 2000; Lucas et al., 2014). Typically these studies are conducted in person. Recently Kosoy et al. (2022) proposed an open-sourced virtual blicket detector environment (Figure 1), focused on studying how children learn and use causal overhypotheses in exploration. This virtual environment has several advantages: it enables not only tracking interactions with children, but also an *exact* comparison between children and machine learning agents via the OpenAI Gym interface (Brockman et al., 2016a). Please refer to Appendix A for screenshots of the virtual blicket detector in various states.

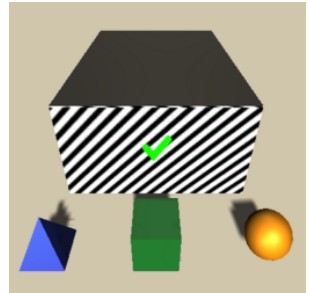

**Figure 1:** A screenshot of the embodied version of the blicket detector.

**Causal Overhypotheses** Causal overhypotheses are hierarchical priors over the structure and/or conditionals of a causal model, and have been widely studied in the cognitive science literature (Kemp et al., 2007; 2010; Lucas & Griffiths, 2010; Perfors et al., 2011; Tenenbaum et al., 2011; Gopnik & Wellman, 2012). An overhypothesis might state that the causal graph itself has a particular form (such as a "common effect" or a chain structure), or that the conditionals within that graph have a particular form (such as that $p(X|Y, Z)$ follows a particular parametric distribution). Having a good causal overhypothesis is a form of inductive bias that can make causal inferences much easier: for example, while we might not know the *specific* causal graph, if we know that it takes the form of a common effect, then we need only a few interventions (possibly only $O(N)$) to fully determine the specific causal graph. Ideally, then, machine learning agents should be able to learn such overhypotheses and leverage them to make more efficient and accurate causal inferences.

In our environment, we consider two causal overhypotheses: that the world is either *conjunctive* or *disjunctive*, with the causal system defined as follows. In both cases, the causal graph takes the form of a common effect, with $N + 1$ variables: $N$ objects $\mathbb{X} = \{X_0, X_1, ..., X_N\}$ (the causes) and one blicket machine $M$ (the effect). Each object $X_i$ can either be on top of the blicket machine ($X_i = 1$) or off the machine ($X_i = 0$). The blicket machine can be either on ($M = 1$) or off ($M = 0$). Because the graph is a common effect, objects' states do not influence each other (intervening on $X_i = 1$ does not impact $X_j$). Objects' states may, however, influence the state of the machine. Specifically, some subset of the objects, $\mathbb{B} \subset \mathbb{X}$, are said to be "blickets" in that they have a causal influence on whether the blicket machine turns on. Thus, the causal graph in this scenario always take the form of a common effect, with edges $\{X_i \rightarrow M : X_i \in \mathbb{B}\}$.

The conjunctive and disjunctive overhypotheses specify the form of the blickets' causal influence on the machine. For example, let $X_i, X_j \in \mathbb{B}$ be blickets. In the conjunctive case, both objects are

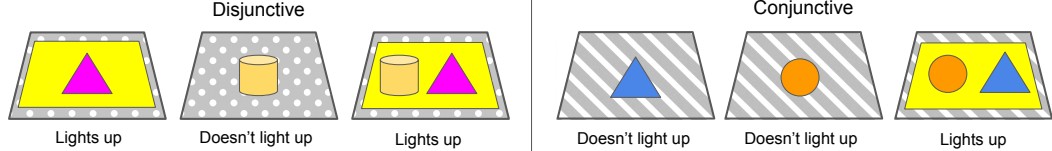

**Figure 2:** A simplified rendering of the virtual blicket detector environment. In the disjunctive setting (left), only one blicket is needed for the blicket to light up. Whereas in the conjuctive setting (left), two blickets are needed. These examples are shown to children in Kosoy et al. (2022) for the *given hypotheses* condition.

needed at the same time to turn the machine on. Formally, $P(M = 1|X_i = 1, X_j = 1) = 1$, while $P(M = 0|X_i = 1) = 1$ and $P(M = 0|X_j = 1) = 1$. In the disjunctive case, only one object (either $X_i$ or $X_j$) is needed to turn the machine on, so $P(M = 1|X_i = 1) = 1$ and $P(M = 1|X_j = 1) = 1$. An illustration of the disjunctive versus conjunctive overhypotheses can be found in Figure 2. We note that while this setup is deterministic, it can easily be adapted to a stochastic environment with minimal modifications.

**Prior Experiments with Children** Since we would like to compare the performance of machine learning agents to that of children in this environment, we aim to make our evaluation of machine learning algorithms comparable to how children are evaluated in Kosoy et al. (2022). Their evaluation used a $2 \times 2$ experimental design, with a *given hypotheses* or *not given hypotheses* training phase, combined with a *disjunctive* or *conjunctive* causal relationship in the test phase. In the training phase, they showed the child two different blicket detectors, with three unique objects each, and for each detector demonstrated three examples of placing objects on the detector and checking whether it lights up. In the *given hypotheses* case, one could infer from the examples that one detector was conjunctive and the other was disjunctive, as shown in Figure 2. Whereas in the *not given hypotheses* case, one could not infer from the examples whether the detectors were conjuctive or disjunctive. In the test phase, the children were given the opportunity to freely interact with a third blicket detector with three new objects. When they were done exploring, they were asked to respond yes or no to which objects were blickets. In their experiments with children from ages four to six, Kosoy et al. (2022) found that children exhibit a diverse range of exploration strategies, which suggests that they are exploring based on a rich set of causal overhypotheses, formed from their prior knowledge of how objects and mechanisms behave.

**Experiments with SOTA RL Models** To adapt the virtual blicket environment for agent learning in this work, we made the following design choices regarding the observations, actions, reward, and termination conditions.

**Observations:** We could allow the algorithms to observe the same embodied visual space as the children, but this places RL algorithms at a significant disadvantage, since they would need to not only understand causal structures, but also learn visual inputs and object detection. Thus, we choose to evaluate the algorithms in a purely symbolic environment where the objects are represented by one-hot indices. Formally, the state space is a vector in $o \in \{0, 1\}^{N+1}$ where $N$ is the number of blickets. The index $o_i, 0 \le i < N$ is 1 if $o_i$ is on the detector, and 0 otherwise. The index $o_N$ is 1 if the detector is illuminated, and 0 otherwise.

**Actions:** In the experiments with children, actions consist of placing a blicket onto the detector, removing a blicket from the detector, and pressing the "check" button to evaluate the detector's state. For RL algorithms, we simplify this process by allowing the agent to place multiple objects simultaneously, automatically "checking" the state of the detector with every action, and automatically resetting the detector after each check, leading to $2^N$ actions for $N$ blickets. This means that the agent gets feedback with every action, which significantly improved training stability.

**Reward:** The reward function should capture whether the algorithm has learned the causal overhypothesis of an environment. To do this, we evaluate the models using a quiz-based framework. Models are allowed to make as many exploration steps as needed, and then trigger an action which switches to the evaluation mode. In the evaluation mode, models receive as input a blicket and must produce an action indicating if the object is a blicket or not. They receive a reward of 1 for identifying a correct blicket, and a reward of $-1$ for incorrectly labeling an object (i.e., for both false positives and false negatives). We also explored several reward modifications. In one modification, models were asked to disambiguate between Disjunctive and Conjunctive environments, with +1 reward for identifying the correct modality. The environment is implemented using the standard OpenAI Gym

(Brockman et al., 2016b) interface, allowing it to be used across many different pre-existing machine learning architectures and algorithms.

## 4 EVALUATING CAUSAL LEARNING IN THE BLICKET ENVIRONMENT

Results from Kosoy et al. (2020) suggest that children can explore efficiently, especially given the causal overhypotheses. In this work, we evaluate how a spectrum of different machine learning models perform on the blicket detector tasks. Solving these tasks requires reasoning about the sequence of observations seen, extracting causal overhypotheses from those observations, and using these extracted overhypotheses for exploration.

We first evaluate several popular reinforcement learning algorithms—A2C (Mnih et al., 2016), PPO2 (Schulman et al., 2017), and Q-learning (Watkins & Dayan, 1992)—on this task. We also evaluate imitation learning algorithms, including behaviour cloning with decision transformers (Chen et al., 2021). Finally, we apply pre-trained language models (Brown et al., 2020; Chowdhery et al., 2022) to this task, since they have been shown to be capable of performing common-sense reasoning and expressing uncertainty to an extent (Lin et al., 2022).

### 4.1 DEEP REINFORCEMENT LEARNING ALGORITHMS

We evaluate the performance of two popular deep reinforcement learning algorithms, Advantage Actor Criric (A2C) (Mnih et al., 2016) and Proximal Policy Optimization (PPO2) (Schulman et al., 2017), on the blicket environment. For each algorithm, we use several policy variants: a standard MLP policy (no memory), an LSTM-based policy, and an LSTM-based policy with layer normalization with two hidden dimension variants of 256 and 512. For all of these policies, we found that a network with a hidden size of 512 obtained the optimal performance See Appendix B for learning hyperparameters. We train all of these algorithms in the *given hypothesis* scenario, where the agent is exposed during training to all possible overhypotheses, and asked to perform well given these scenarios.

**Experimental Design** We terminate training either after 3 million environment steps or when the agent obtains maximum reward for 500 consecutive episodes, whichever comes first. Each episode has 25 timesteps, so each agent is exposed to at most 120,000 episodes. Experiments are run using a single Nvidia Titan-X Maxwell GPU, and take less than three hours to complete. To evaluate whether the agent can generalize to additional causal situations, we also train agents on five held-out scenarios: holding out all of the conjunctive overhypotheses, holding out all of the disjunctive overhypotheses, and holding out either one conjunctive overhypothesis or one disjunctive overhypothesis. The maximum achievable reward for these experiments is 3.

**Results** Figure 3a shows the performance of A2C and PPO2 when none of the hypotheses are held out. PPO2 outperforms A2C in almost all scenarios, achieving higher rewards faster. Further, the LSTM models clearly outperform the non memory-based models; this is expected, since causal learning requires memory. Unfortunately, none of the algorithms perform well on held-out causal examples, as shown in Figure 3c. This suggests that they are primarily learning to memorize the causal patterns, and thus are incapable of generalization. This conclusion comes with a caveat: because there are only a handful of possible hypotheses, it may be possible that we do not have enough data to perform well on held-out samples. Agents are, however, able to easily distinguish between conjunctive and disjunctive environments—Figure 3b shows that after very few steps, in the held-out situation (with both overhypotheses), agents can distinguish conjunctive from disjunctive environments, even though they are unable to determine which objects exactly are blickets.

### 4.2 Q-LEARNING

We also train tabular Q-Learning on the symbolic version of the blicket environment. We append the full history of previous observations to the current observation in order to give the agent memory. The Q-values are initialized to zero. We used $\epsilon$-greedy exploration with an exploration probability of $0.1$, and we found the best learning rate was $0.95$. Q-learning is able to learn the task very quickly due to a small search space—it took an average of 70 episodes and 292 steps for the agent to converge to maximum reward. However, tabular Q-learning is incapable of generalizing to new scenarios, so we do not test Q-Learning agents in the held-out scenarios.

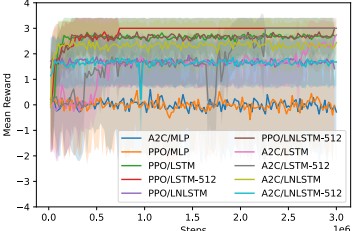

**(a)** Performance of PPO2 and A2C algorithms with MLP, LSTM, and Layer-Norm LSTM policies on the blicket environment.

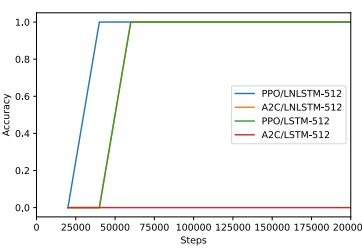

**(b)** Accuracy of models over time when determining if the environment is conjunctive or disjunctive.

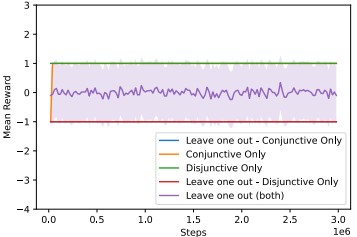

**(c)** Performance on several over-hypothesis variants. Conjunctive Only: Train on conjunctive hypotheses, test on conjunctive and disjunctive. Disjunctive Only: Train on only disjunctive hypotheses, test on conjunctive and disjunctive. Leave one out modifier: Train on 2/3 (or 4/6 for both) of the hypothesis within the overhypothesis, and test on the remaining.

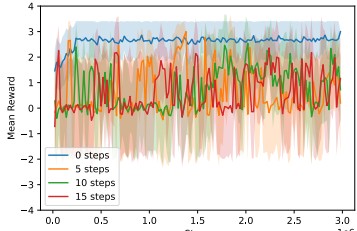

**(d)** Performance of a standard A2C-LSTM model when forced to explore for K steps before entering the quiz phase. While this exploration helps the decision transformer, additional forced explanation is harmful to all standard RL models.

**Figure 3:** Experiments with standard deep reinforcement learning algorithms

| Model | Pre-Training Dataset | Reward | FCA |
|---|---|---|---|
| Decision Transformer | Random | $0.16 \pm 1.747$ | 0.67 |
| Decision Transformer | Random (5-step forced exploration) | $2.06 \pm 0.998$ | 0.71 |
| Decision Transformer | Random (10-step forced exploration) | $2.16 \pm 0.872$ | 0.73 |
| Decision Transformer | Random (15-step forced exploration) | $2.37 \pm 0.633$ | 0.77 |
| Decision Transformer | A2C-LSTM | $0.14 \pm 1.847$ | 0.63 |
| Decision Transformer | PPO2-LSTM | $0.26 \pm 0.990$ | 0.66 |
| Decision Transformer | A2C-LayerNorm-LSTM | $0.18 \pm 0.983$ | 0.62 |
| Decision Transformer | PPO2-LayerNorm-LSTM | $0.74 \pm 1.460$ | 0.64 |
| Behavior Cloning | Random | $0.22 \pm 0.975$ | 0.61 |
| Behavior Cloning | Random (5-step forced exploration) | $-0.16 \pm 0.987$ | 0.58 |
| Behavior Cloning | Random (10-step forced exploration) | $0.14 \pm 0.990$ | 0.60 |
| Behavior Cloning | Random (15-step forced exploration) | $0.04 \pm 0.999$ | 0.61 |
| Behavior Cloning | A2C-LSTM | $0.10 \pm 0.995$ | 0.55 |
| Behavior Cloning | PPO2-LSTM | $0.06 \pm 0.998$ | 0.63 |
| Behavior Cloning | A2C-LayerNorm-LSTM | $0.04 \pm 2.087$ | 0.59 |
| Behavior Cloning | PPO2-LayerNorm-LSTM | $0.12 \pm 0.995$ | 0.68 |

**Table 1:** Performance of the imitation learning models with different pre-training datasets. Reward is on the blicket-quiz task: +1 for correctly identifying a blicket, and -1 for incorrectly identifying a blicket. FCA refers to Forced Choice Accuracy, accuracy of the model in determining if the environment is conjunctive or disjunctive.

### 4.3 Behavior Cloning

Recently Chen et al. (2021) introduced the Decision Transformer, a simple transformer-based approach to imitation learning, shown to outperform most existing behavior cloning methods. The decision transformer works by applying a causally-masked transformer to predict the reward-to-go of a flattened sequence of (state, action, reward, next state) tuples, with an $\ell^2$-norm loss. The model then chooses actions during test time that maximize the predicted reward-to-go.

**Experimental Design** We collect trajectories for behavior cloning by exploring randomly in the space (an approach followed by Chen et al. (2021)). In this work, we adapt the model to predict discrete actions in our space by adding a sigmoid activation to the action predictions and altering the action prediction loss accordingly. We evaluate the model using a target reward of 3, which corresponds to identifying all of the blickets correctly. The decision transformer was trained using a batch size of 128, a K-value of 30, embedding dimension of 128, 3 layers, one head, and dropout of 0.1. The weights are optimized using ADAM, with a learning rate of $1e^{-4}$ and weight decay of $1e^{-4}$. We also compare against standard behavior cloning, using an MLP with a hidden-dimension of 128, and trained using the same optimizer as the decision transformer. Both models are trained for 10 epochs using one Nvidia Titan-X Maxwell GPU, which takes less than one hour, and the best validation set checkpoint is used for each during test time. Models are evaluated on 100 rollouts in the environment, and the mean reward is reported.

When training the decision transformer model, we found that in some cases, random (and even the PPO2/A2C trained) models were unable to explore efficiently, as they entered the quiz environment too soon. Note that random exploration will enter the quiz environment after $t$ steps with with probability $p = 1 - \frac{1}{2^t}$. Thus, we found it helpful to force the policy to explore for several steps before allowing it to enter the quiz environment. While this process helps the decision transformer, it negatively affects the performance of the standard RL models, as shown in Figure 3d.

**Results** Table 1 shows the performance of the decision transformer and behavior cloning models when applied to datasets generated by several policies. As we can see, while expert policies allow for higher rewards using the decision transformer, allowing the model additional forced exploration time is the most important factor. This suggests that on their own, the A2C and PPO2 trained policies do not lead to sufficient exploration for learning a strong model of the reward, whereas forcing additional exploration (even if it is random) is much more useful. Notably, standard behavior cloning performs very poorly, as copying actions with the same local observations under different overhypotheses will likely lead to incorrect or uninformative actions.

### 4.4 Causal Discovery Baselines

Most deep learning models do not explicitly model the causal structure of the data, for example, it would be quite challenging to extract the causal structure learned by a transformer network. However, one workaround is to evaluate the model on causal structure learning by asking it to answer questions about the existence of an edge in a causal graph. For example, we could ask a language model, does variable A cause variable B? Then one can take the average accuracy of these answers as the ratio of the structured hamming distance (SHD). However, note that in our case, since objects do not influence each other, we only ask the question of which objects influence the machine.

To explore the performance of existing causal discovery models on the benchmark, we leveraged the recent Differentiable Causal Discovery from Interventional Data (DCDI) approach (Brouillard et al., 2020). We evaluated the model on causal structure learning for the disjunctive and the conjunctive settings. For the disjunctive settings, the model achieved a structured hamming distance (SHD) of 2.3. Noting that the graph has 4 variables and hence 16 possible edges in the graph structure, the model achieved a SHD ratio of 85%. For the conjunctive setting, the model achieved a SHD of 2.6, which is a SHD ratio of 83%.

### 4.5 Large Language Models

Recently large language models have shown promising performance on a wide variety of tasks, including logical inference and common-sense reasoning. Such language models include GPT-3 (Brown et al., 2020), PaLM (Chowdhery et al., 2022), MT-NLG (Smith et al., 2022), and Chinchilla (Hoffmann et al., 2022), which all use transformer-based architectures and are trained autoregressively on a large corpus of text, to predict the next token when given a sequence of tokens. These language

models have also shown promising performance on tasks that involve causal reasoning, such as identifying causal effects from observational data (Veitch et al., 2020) and inferring causality for when drugs lead to adverse effects (Wang et al., 2021b). However, these models have not yet been applied to a task that involves reasoning across a set of overhypotheses, as is required in the blicket environment.

Motivated by this, here we investigate the performance of two such language models, GPT-3 and PaLM, on the blicket environment. One can apply a language model to a target task via purely text-based interaction with the model, by providing a prompt to the model and evaluating its output. This prompt could be *freeform*, in which it has no set structure, or it could be *few-shot*, where it contains a small number of examples of correct prompt-output pairs, followed by the test prompt. Instead of text-based interaction, another option is to fine-tune the model on the target task. This involves training the model on data from the target task, and can require a sizeable amount of data. To make the evaluation more similar to how children are introduced to the task, we focus on only text-based interaction with the model, including both freeform and few-shot prompts.

As in the experiments with children, there are four conditions: *given* versus *not given hypotheses*, and *disjunctive* versus *conjunctive*. In the *disjunctive* case, one of the three objects is a blicket. In the *conjunctive* case, two of the three objects are blickets. We evaluate how well the models can: 1) identify the blickets and 2) identify whether the causal structure is disjunctive or conjuctive. For GPT-3, we use the OpenAI API[1], and for PaLM we use an internally available interface. We have also run GPT-3 on different temperature settings and we found that this did not significantly change the result, refer to Section C.1 in the appendix for details.

**Freeform Prompt** In the freeform prompt, we provide the model with text that is as similar as possible to what children receive in the blicket experiment. The only difference is that we replace the visual components with text descriptions. We include the exact prompts in Appendix C. The prompt starts with a description of interacting with a striped blicket machine and then a dotted blicket machine, each of which has three unique objects. There are three interaction examples given per machine, for example "If we put the blue pyramid on the machine, then it does not light up". In the *given hypotheses* condition, the striped blicket machine is conjunctive and the dotted blicket machine is disjunctive. In the *not given hypotheses* condition, there is not enough information to determine whether the striped and dotted blicket machines are disjunctive or conjunctive. Thus the *given hypotheses* condition defines the space of overhypotheses, whereas the *not given hypotheses* condition does not. The prompt then introduces a new blicket machine, along with examples of interactions with the machine and whether it lights up or not. Finally we ask the model which objects are blickets and whether the new machine is more similar to the striped or dotted machine.

**Few-shot Prompt** In the few-shot prompt, we structure the input into two prompt-output demonstrations, one per machine, containing the same information as in the freeform prompt. Unlike the information given to children, in the outputs we explicitly state which objects are blickets. In addition, for the *given hypotheses* condition, in the preamble we define the disjunctive and conjunctive hypotheses: "A striped machine needs two blickets to make it light up, and a dotted machine needs one blicket to make it light up", and in the outputs we state whether this is more similar to a striped or dotted machine.

The results are reported in Table 2.[2] We find that when given hypotheses, GPT-3 and PaLM are almost always able to select the correct causal structure, but they are not always able to select the correct blickets. In particular, GPT-3 frequently names too many objects as blickets, including those associated with other machines. In contrast, PaLM never identified objects as blickets that were not one of the three objects in the test task. In the two-shot setting, PaLM performs best when the space of overhypotheses is covered perfectly by the two examples given, as one would expect. However, when this space is not covered, i.e. in the *not given hypothesis* setting, PaLM struggles in the two-shot setting because the test example's causal structure does not match either of the two given examples. Adding chain-of-thought reasoning in the PaLM prompt did not improve results in either setting.

---

[1]Available at https://beta.openai.com/overview

[2]For GPT-3, we use the text-davinci-002 model, with temperature 0.7, maximum length 256, and frequency and presence penalties 0. For PaLM, we use a temperature of 0 to obtain the greedy, or one-best decoding. When given the freeform input, the PaLM model output continues indefinitely, so we simply took the first two sentences of the output and ignored the rest.

| Condition | Model & Input | Blickets Chosen | Causal Structure |
|---|---|---|---|
| Given hypotheses, disjunctive | GPT-3, freeform | 1/1 correct, 6 wrong | **correct** |
| | PaLM, freeform | 1/1 correct, 1 wrong | wrong |
| | PaLM, two-shot | **1/1 correct** | **correct** |
| Given hypotheses, conjunctive | GPT-3, freeform | 2/2 correct, 1 wrong | **correct** |
| | PaLM, freeform | **2/2 correct** | **correct** |
| | PaLM, two-shot | **2/2 correct** | **correct** |
| Not given hypotheses, disjunctive | GPT-3, freeform | 1/1 correct, 7 wrong | — |
| | PaLM, freeform | 0/1 correct | — |
| | PaLM, two-shot | 1/2 correct, 1 wrong | — |
| Not given hypotheses, conjunctive | GPT-3, freeform | 2/2 correct, 7 wrong | — |
| | PaLM, freeform | **2/2 correct** | — |
| | PaLM, two-shot | 1/2 correct, 1 wrong | — |

**Table 2:** Results for GPT-3 and PaLM for the four conditions. Bold font indicates a fully correct answer. For the *not given hypotheses* setting, answering correctly is impossible, as there is not enough information to determine whether the blicket detectors in the training phase are disjunctive or conjunctive.

## 5 DISCUSSION & CONCLUSION

In this work, we looked into evaluating and understanding how machine learning models learn causal overhypotheses, by evaluating these models in the blicket environment. In contrast to existing benchmark tasks, in which there is a fixed causal structure, this environment focuses on the need for *causal overhypotheses* in order to explore effectively to determine the underlying causal structure. We focused on three categories of state-of-the-art methods—deep RL, behavior cloning, and large language models—for tasks in this environment. In blicket detector experiments, children are able to learn causal overhypotheses from only a handful of observations and can apply these overhypotheses to explore effectively for a new situation (Kosoy et al., 2022). In contrast, our experiments indicate that state-of-the-art machine learning algorithms have difficulty learning and using causal overhypotheses for exploration and inference—we saw this in the weak performance of deep RL algorithms on held-out environments and in the tendency of decision transformer models to under-explore. In our experiments with language models, we provide the same observations that the children were given in Kosoy et al. (2022), along with a full set of examples for the new situation (thus removing the need for exploration). Despite this, language models struggle when the hypotheses are not given, and are not able to express uncertainty about the causal structure in that case.

Given that understanding and leveraging causal structure is essential to developing general intelligence, this work highlights an opportunity for improvement in this direction, and provides a set of concrete benchmark tasks to measure improvement. One direction of future work is to build machine learning models that can better learning causal overhypotheses. Modular architectures have shown to be helpful in understanding causal hypotheses of the environment (Goyal et al., 2019; 2021; Ke et al., 2021); it would be promising to explore such models for causal overhypotheses understanding. Another direction of future work is to improve on exploration in RL agents by explicitly learning and incorporating causal overhypotheses, in order to narrow down the search over possibilities. Furthermore, an interesting direction is to train machine learning models on children's exploration behavior used in solving causal problems, for instance the trajectories provided by Kosoy et al. (2020). This may lead to new insights for improving causal overhypotheses learning for machine learning models.

A main limitation of this work is that we evaluated only two types of causal overhypotheses–conjunctive and disjunctive. These are particular types of overhypotheses on the causal graph structure. In the real world, there exist many other types of overhpotheses, as well as conditional probability distributions. It would be interesting to see if one can extend either the blicket or other RL environments (such as Ahmed et al. (2020); Ke et al. (2021)) to include other types of causal overhypotheses. With minor changes, one could extend the blicket environment to test other kinds of causal overhypotheses, such as inferring whether systems are stochastic or deterministic, or require sequential or unordered interventions. This environment also allows for multiple measures of causal inference, including interventions and counterfactuals, as well as predictions.

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
