# OpenReview forum: "Towards Understanding How Machines Can Learn Causal Overhypotheses "
_ICLR.cc/2023/Conference — Submitted to ICLR 2023_

### Official Review · Reviewer_ypFh · 2022-10-26

**Confidence:** 3
**Correctness:** 2
**Technical Novelty And Significance:** 3
**Empirical Novelty And Significance:** 3
**Recommendation:** 5

**Clarity, Quality, Novelty And Reproducibility:**

I do not believe that the experiments are reproducible based on the current description in the paper. A lot of important moments (see weaknesses) are unclear. The main idea, learning priors for overhyposes to guide exploration, seems novel to me.

**Strength And Weaknesses:**

Strength
1) The paper is well-motivated;
2) Results with LLM  are very interesting;
3) A direct comparison of any algorithms with children of age four can be made.

Weaknesses
1) My main concern is that I did not understand how the proposed benchmark test for overhypotheses. I do not understand how one can understand that a considered algorithm or a child learned causal overhyposes. Can you elaborate on it? The authors wrote “The reward function should capture whether the algorithm has learned the causal overhypothesis of an environment”, but  I do see only that the proposed reward can help distinguish whether the agent learns to discover (or generalize to discover) a causal graph (or blinkets) during the test/eval stage.
2) Following the 1. The authors wrote: “We find that although most agents have no problem learning causal structures seen during training, they are unable to learn causal overhypotheses from these experiences, and thus cannot generalize to new settings.” I do see that algorithms did not generalize to the new settings, but I do not understand why it is due to overhypotheses. In some sense, I feel the opposite. For example, in the case when models are trained only on disjunctive data, the model learns from the data that conjunctive case is not possible. So the model learns overhyposes prior perfectly. Another story is that the model encoded this prior to the weights of the model and can not recover from it and generalize to conjunctive out-of-distribution data in the test stage.
3) I doubt that any reliable conclusion can be made based on such a small hold-out test set. I believe the experimental results rather motivate father research in this direction than allow to make reliable conclusions.  (the authors noted it themself). Maybe the easiest workaround is to consider more than three objects and more than two blinkets to train and test algorithms;
4) I do not understand how the author trains RL algorithms. For example, the author wrote: “We train all of these algorithms in the given hypothesis scenario, where the agent is exposed during training to all possible overhypotheses”.  I do not understand what this means. "Given hypothesis" is a case where an agent receives demonstration data, but how this offline data is used to train RL agent? Is it used in offline RL settings? Do the authors train the agent on-policy then?


Small issues and questions
1) If the maximum achievable reward is always 3, I believe the authors give a reward for correctly identifying the identity of the object (is it a blicket or not). The author's wording “they receive a reward of 1 for identifying a correct blicket” confuses me to think that the maximum reward is 1 for the disjunctive setting and 2 for the conjunctive setting.
2) What is the distribution of the tasks when you train RL agent?
3) In the situation when you hold out disjunctive overhypothesis, does it mean that the training set does not contain examples when object 3 (without loss of generality) is a blanket?
4)  If the agent allows checking all the options for the blinkeness at the same time, is it one of the working strategies for the ideal algorithm to check all combinations in one round (it is only 8 for 3 objects) and then conclude what objects are blinkets?


I also believe the paper is strongly connected to meta-learning in RL, in which the agent needs first explore an environment to understand what task it should solve and then solve this task (Rakelly et al, Efficient Off-Policy Meta-Reinforcement Learning via Probabilistic Context Variables, ICML2019).

**Summary Of The Paper:**

The paper presents a causal learning benchmark to test the learning of causal overhypotheses. The authors test several RL algorithms as well as pretrained language models on this benchmark.

**Summary Of The Review:**

I think the main idea of the paper is novel and attractive. However, I need a lot of clarifications to properly evaluate the paper.

---

### Official Review · Reviewer_cvDM · 2022-10-27

**Confidence:** 3
**Correctness:** 3
**Technical Novelty And Significance:** 3
**Empirical Novelty And Significance:** 3
**Recommendation:** 3

**Clarity, Quality, Novelty And Reproducibility:**

Novelty: Good
Reproducibility: Not sure

**Strength And Weaknesses:**

Strength:
I think the paper is quite novel in trying to bridge other non-computational disciplines to ICLR.

Weakness:
If I understood it correctly, the paper seems to be only running some experiments and check some performances which has an association with the causal measures on the over-hypothesis. But the paper did not give any insight. It looks to be providing a benchmark suite. In the RL experiment, the visual image is replaced with symbolic representation also.

**Summary Of The Paper:**

The paper utilized a recently developed framework of "blicket" environment and conducted experiments using reinforcement learning, imitation learning and Language models. The goal of the paper is to understand if these machine learning method could acquire a causal understanding of the combinations of objectives to the "blicket".



**Summary Of The Review:**

Learning causal over-hypothesis is essential for AI to robustly help us to deal with automated tasks under the challenge of distribution shift. If my understanding were correct, the paper proposed a benchmark suite, which I am not fully convinced about the importance of this suite compared to constructing some other benchmark to evaluate the causal performances of machine learning algorithms.

---

### Official Review · Reviewer_6AJY · 2022-10-30

**Confidence:** 3
**Correctness:** 3
**Technical Novelty And Significance:** 2
**Empirical Novelty And Significance:** 3
**Recommendation:** 6

**Clarity, Quality, Novelty And Reproducibility:**

This work is a continuation of (Kosoy et al. 2022) (by the way, some remarks in the paper strongly suggest that it is probably the same team's work, which should have been avoided for the sake of blind review), but since very little has been written on this important topic, I think the originality is considerable. The paper is mostly clear, but as I said above, the implications of the experimental results are not always clearly delineated. In particular, it does not seem to be entirely fair to suggest that these machine learning models are not good at learning the overhypotheses considered in this paper. The difficulty seems to have more to do with using the overhypotheses.

As an empirical study, the overall quality is reasonably high. I also believe the reproducibility will be reasonably good.

Minor comment: it is unclear to me what roles are played by the baselines described in Section 4.4.

**Strength And Weaknesses:**

Strengths:

1. Learning causal overhypotheses is an important task to be addressed in machine learning.
2. A variety of machine learning models are considered and assessed.
3. The experimental results are mostly interesting and telling, and some potentially useful benchmarks are given.

Weaknesses:

1. The distinction between learning an overhypothesis and exploiting the overhypothesis in exploration or inference of causal relations is not always clearly made. As I understand it, the target overhypothesis in this paper concerns whether the causal mechanism is conjunctive or disjunctive. If so, the results in the paper seem to suggest that at least the RL agents and the language models are able to learn the correct overhypothesis most of the time; what they seem to be unable to do well in the blicket environment is take advantage of the overhypothesis to figure out which objects are blickets. Is this reading correct? If it is, it is not clearly described or explained in the paper.

2. The work is purely empirical and limited to certain ways to apply reinforcement learning, imitation learning, and large language models to causal inference. It is unclear whether the observed weaknesses are due to these approaches or to the authors' particular implementations of these approaches for causal inference.

**Summary Of The Paper:**

This paper leverages the (virtual) blicket detector environment to empirically evaluate the ability of machine learning models to learn and use causal overhypotheses, which refer to those hypotheses that concern not specific causal relations but more general constraints about causal relations, such as whether a causal mechanism is conjunctive or disjunctive. This work is motivated by solid findings in the cognitive science literature on the competence of young children to learn causal overhypotheses and exploit them to probe causal relations in this environment. It builds on the setup in (Kosoy et al. 2022) and designs interesting experiments to assess a range of machine learning models regarding the learning of overhypotheses, including popular deep reinforcement learning algorithms, some imitation learning algorithms, and some large language models. Various weaknesses of these machine learning agents are empirically demonstrated.

**Summary Of The Review:**

It is a useful empirical study about an important topic that should receive more attention in machine learning.

---

### Decision · Program_Chairs · 2023-01-20

**Decision:**

Reject

**Justification For Why Not Higher Score:**

The work is purely empirical, and several issues should be further addressed (e.g., regarding the distinction between learning an overhypothessi and exploiting the overhypothesis). I wish the authors had provided responses to the reviews, given that most reviews have good questions in their reviews.

**Justification For Why Not Lower Score:**

The studied problem, learning causal overhypotheses, is an important task in machine learning, and the proposed approach is based on solid findings in cognitive science.

**Metareview: Summary, Strengths And Weaknesses:**

This paper leverages the virtual blicket detector environment to empirically evaluate the ability of machine learning models to learn and use causal overhypotheses. The causal overhypotheses concern not specific causal relations but more general constraints about causal relations, such as whether a causal mechanism is conjunctive or disjunctive.  The paper conducted experiments using reinforcement learning, imitation learning, and Language models. The studied problem, learning causal overhypotheses, is an important task in machine learning, and the proposed approach is based on solid findings in cognitive science. However, the work is purely empirical, and several issues should be further addressed (e.g., regarding the distinction between learning an overhypothessi and exploiting the overhypothesis). I wish the authors had provided responses to the reviews, given that most reviews have good questions in their reviews.